# Dough Rheological Behavior and Bread Quality as Affected by Addition of Soybean Flour in a Germinated Form

**DOI:** 10.3390/foods12061316

**Published:** 2023-03-20

**Authors:** Denisa Atudorei, Silvia Mironeasa, Georgiana Gabriela Codină

**Affiliations:** Faculty of Food Engineering, Stefan cel Mare University of Suceava, 720229 Suceava, Romania

**Keywords:** germinated soybean flour, refined wheat flour, fundamental rheology, dough microstructure, bread quality

## Abstract

This study analyzes the possibility of using soybeans as an addition to the main ingredients used to make bread, with the aim of improving its quality characteristics. To maximize the nutritional profile of soybeans they were subjected to the germination and lyophilization process before being used in bread making. The addition levels of 5%, 10%, 15%, and 20% germinated soybean flour (GSF) on dough rheology and bread quality were used. From the rheology point of view, the GSF addition had the effect of decreasing the values of the creep and recovery parameters: J_Co_, J_Cm_, μ_Co_, J_max_, J_Ro_, J_Rm_, and J_r_. At the same time, the rheological parameters λ_C_ and λ_R_ increased. The GSF addition did not affect dough homogeneity as may be seen from EFLM analysis. Regarding the quality of the bread, it may be concluded that a maximum of 15% GSF addition in wheat flour had a desirable effect on loaf volume, porosity, elasticity, and sensory properties of the bread. The bread samples with GSF additions showed a higher brightness and a less pronounced red and yellow tint. When the percentage of GSF in wheat flour increased, the value of the firmness parameter increased and the value of the gumminess, cohesiveness, and resilience parameters decreased. The addition of GSF had a desirable influence on the crumb structure of the bread samples. Thus, taking into account the results of the determinations outlined above, it can be stated that GSF addition in wheat flour leads to bread samples with good quality characteristics.

## 1. Introduction

In today’s society, consumers are concerned about adopting a balanced lifestyle by limiting the use of food additives of chemical origin [1] and using natural products in foods [2], which will bring a balanced nutritional intake between the basic nutritional compounds: proteins, lipids, carbohydrates, vitamins, minerals. As a result, the attention of specialists has recently been directed towards to the production of food products that correspond to both the sensory and nutritional requirements of consumers.

In the modern era, increasing interest is being awarded to the consumption of vegetables such as legumes [3]. This is due to the balanced nutritional profile that these have. Furthermore, following the innovation trend, legumes in germinated form have grown more popular [4] and are incorporated in various forms in food products. From a scientific point of view, this is well received because germination has a very good imprint on the profile of the pulses. Some scientific papers have already concluded that sprouting produces the following effects: nutritional improvement of the grains that were germinated [5], increasing the availability of some nutritional compounds, reducing the amount of anti-nutritional factors [6], increasing the amount of some constituents that were previously found in bound form, improving the sensory profile of the grains [7], etc. Moreover, in the bakery industry, sprouts are of interest due to their high enzymatic activity because germination produces the activation of enzymes specific to the grains [8].

Nutritionally, soybeans contain a high amount of protein, essential vitamins and minerals, complex carbohydrates, and dietary fiber [9]. Soybeans also contain phytochemicals that contribute to the health of the human body [10]. The significant isoflavone content of soybeans is of interest from the point of view of human health. Studies have shown that they have a certain implication in reducing the incidence of some types of cancer, osteoporosis, cardiovascular diseases, and also have a role in tempering the specific symptoms of menopause [11,12].

Over time, soybeans have grown popular on the consumer’s plate as a functional ingredient in various recipes for the preparation of food, resulting in nutritional improvement. However, studies have also closely followed the impact of their incorporation on the qualitative value of food products and from other points of view, not just nutritionally speaking. In the bakery industry, several studies have been conducted on the use of soybean flour in wheat flour dough and its addition to the finished products. Most studies consider the use of using soybean derivatives as an addition in the bread formula, such as soy protein isolate [13], hydrolyzed soy protein [14], soymilk powder [15], fermented soy [16], and soybean polysaccharides [17]. The fact that there are so many studies focused on the possibility of using soybeans in different forms in bread recipes demonstrates the potential of using this valuable legume in bakery products. The utilization of germinated soybeans in the recipe for bread making is less studied. In some scientific papers it was concluded that using germination in a controlled way on grains can have a positive role in trying to make bread with better volume, texture, and sensory properties [18]. From an organoleptic point of view, Dhingra et al. (2002) concluded that the incorporation of 10% soybean flour to wheat flour produced acceptable breads [19]. In general, studies have outlined that supplementing with flour obtained from legumes modifies both the dough rheology and bread quality [20]. Previously, it was established that the addition of a mixture of germinated legume flours from soybeans and lentils caused a reduction in the value of the dough rheological parameters of extensibility, consistency, falling number, tolerance to kneading, and initial gelatinization temperature. It was also demonstrated that the addition of this mixture coincided with an increase in the dough rheological parameters, namely: the maximum height of gaseous production, total CO_2_ volume production, the volume of the gas retained in the dough at the end of the test, dough tenacity, maximum gelatinization temperatures, and viscoelastic moduli [21].

The present study focuses on valuing the implications of the addition of germinated soybean flour, obtained using the lyophilization process to decrease the moisture content of the sprouts, from the point of view of the dough rheology, microstructure, and the qualitative value of the wheat bread. To our knowledge, there are no detailed studies that consider the use of germinated soybean flour as an addition in the bread making recipe, considering both dough microstructure and rheology and also from the bread quality point of view. The present study is welcome considering the superior nutritional quality of soybeans and the advantages brought by their germination, technologically speaking.

## 2. Materials and Methods

### 2.1. Materials

To conduct the present experimental study, the following materials were used: white wheat flour (type 650), germinated soybean flour, compressed yeast *Saccharomyces cerevisiae* type, sodium chloride salt, and water. The purchase of white wheat flour was made from the company S.C. Dizing S.R.L., which is located in Brusturi, Neamț, Romania. Germinated soybean flour (GSF) was obtained by subjecting soybeans (*Glycine max* L.) to the germination and lyophilization process under controlled conditions. The soybeans were germinated respecting some specific conditions. The temperature of the germination process was 25 °C. The humidity during the whole process was 80%. Germination took place over 4 days in exclusive conditions of darkness. The seed germination is described in detail in a separate paper of ours [22]. At the end of the 4 days of germination, the sprouts obtained were lyophilized using a Biobase BK-FD12 lyophilizer. The freeze-drying process was carried out at −50 °C over 24 h at a pressure of 10 Pa. To obtain the germinated soybean flour, the sprouts obtained were ground using a laboratory mill.

To highlight the specific properties of the flours used and the bread chemical composition, the following determinations were made, taking into account the ICC standard methods [23]: ash content (ICC 104/1), moisture content (ICC 110/1), fat content (ICC 136), and protein content (ICC 105/2). The carbohydrate content was determined by difference: 100 − (protein + ash + fat + moisture). Additionally, for wheat flour, the following characteristics were determined: the gluten deformation index (Romanian standard SR 90/2007), the wet gluten content (ISO 21415-2:2015), and the falling number value (ICC 107/1 method).

### 2.2. Dough Fundamental Rheological Properties

To highlight the impact of GSF supplementation on dough rheology, creep and recovery tests were undertaken. The two tests were performed using the HAAKE MARS 40 rheometer (Thermo-HAAKE, Karlsruhe, Germany) at a temperature of 25 °C. A non-serrated parallel plate geometry was used. The diameter was 40 mm and the gap width was 2 mm. For preparation of the dough samples we took into account the optimum dough water absorption, which was determined with an Alveo Consistograph device. The dough samples did not contain yeast. The prepared samples were placed between the rheometer plates and were rested for 5 min. The determinations were performed in the range of linear viscoelasticity at constant stress of 50 Pa in a frequency sweep from 1 to 20 Hz. The creep and recovery tests had a time of 60 s and 120 s, respectively [24].

### 2.3. Dough Microstructure

The changes in the dough microstructure due to the addition of GSF were highlighted performing specific determinations. These were made with the help of a Motic AE 31 (Motic, Optic Industrial Group, Xiamen, China), which was equipped with LWD PH 203 catadioptric objectives (N.A. 0.4). For these determinations, after preparing the samples and considering the method detailed in our previous study [25], a dough piece was cut from each sample. This was introduced for at least 1 h in a fixing solution consisting of 1% rhodamine B and 0.5% fluorescein (FITC) in 2-methoxyethanol. The 2 substances had the role of specific fluorescent dyes, fluorescein to detect starch and rhodamine B for proteins. After placing the samples in the immersion solution, they were examined with the help of ImageJ software (v. 1.45, National Institutes of Health, Bethesda, MD, USA) [24].

### 2.4. Bread Making

The process of bread making followed the main technological stages: proportioning of the raw materials, mixing them, division of the dough, fermenting it, and baking the samples. The ingredients used were: white wheat flour (type 650), GSF (in different proportions: 5%, 10%, 15%, and 20%), 1.5% salt (NaCl), 3% *Saccharomyces cerevisiae* leavening agent (compressed yeast), and water, depending the optimum value of water absorption capacity of flour mixes (white wheat flour and germinated soybean flour). The amounts of water used were: 54.3% for control sample, 54.0% for GSF_5, 53.7% for GSF_10, 53.4% for GSF_15, and 52.5% for GSF_20. After dosing the ingredients, these were mixed for 15 min using a KitchenAid mixer. The samples of dough at 400 g each were then obtained, and these were fermented in a chamber for fermentation. The parameters for fermentation of the samples were: time of 60 min, relative humidity of 85%, and temperature of 30 °C. We used an electrical bakery oven with convection (PF8004 D, Piron, Italy) to bake the samples, which was equipped with steam production, ventilation, and humidification systems. The baking parameters of the bread samples were: temperature of 220 °C and time of 30 min. During the first and last 2 min of baking, we used the steam system with which the oven was equipped [26].

### 2.5. Bread Quality Evaluation

#### 2.5.1. Bread Physical Characteristics

For the determination of the specific volume of the bread samples we used the rapeseed displacement method (AACC Method 10–05.01) [27]. In order to evaluate the porosity and the elasticity, the SR 91:2007 standard method was considered [28].

#### 2.5.2. Color Parameters

We used a colorimeter to highlight the changes in the color of the bread crust and crumb after the addition of GSF (Konica Minolta CR-400, Tokyo, Japan). For this purpose, the darkness/brightness (*L**), shade of blue/yellow (*b**), and shade of red/green (*a**) were determined. The calibration of the colorimeter was achieved by scanning the standard white surface calibration plate (L* = 97.63, a* = 0.01 and b* = 1.64). For this, the standard illuminant D65 (working at daylight) and a 0° viewing angle was used. The determinations were made in the field of UV-VIS based on the CIE Lab* color system [26].

#### 2.5.3. Texture Profile Analysis

For evaluation of the changes in the textural profile of the bread due to the addition of GSF, a texturometer device TVT-6700 (Perten Instruments, Hägersten, Sweden) was used. We cut 50 mm high slices from the bread for analysis. The texturometer was equipped with a 10-kg load cell. The slices of bread were compressed twice, up to 20% of their initial height. The trigger force used was 5 g, the speed was 1.0 mm/s, and the recovery period between compressions was 15 s [29].

#### 2.5.4. Crumb Structure

To observe the crumb structure of the bread, the Motic SMZ-140 stereo microscope (Motic, Xiamen, China) was used with a 20× objective [29]. The images acquired at a resolution of 1024 × 768 pixels were processed with ImageJ software (ImageJ 1.53 version, National Institutes of Health, USA) following the method described by Iuga et al. [30]. As gas cells, the shapes larger than 0.01 mm^2^ were considered because the human eye can perceive particles of approximately 0.1 mm [31].

#### 2.5.5. Sensory Analysis

For the organoleptic analysis test of the bread by the tasters participating in the tasting session, we used a 9-point hedonic scale from 1 to 9 (1 = dislike extremely, 5 = neither like nor dislike, and 9 = like extremely). Twenty semi-trained individuals participated in the organoleptic analysis test by tasting the samples. The sensory analysis methodology used in this study was approved by the Bioethics Committee of our faculty department. The individuals who participated in this study were informed of the aims, protocols, and methodology of the study and gave their consent to participate.

### 2.6. Statistical Analysis

The statistical package for social science (v.16, SPSS, Chicago, IL, USA) was used for the statistical processing of the data. This package helped to establish the statistical significance of the data. All data were expressed as the mean ± standard deviation. A 1-way analysis of variance (ANOVA) and the Tukey’s test were used at a level of 5% of significant differences [26].

## 3. Results

### 3.1. Flour Characteristics

The characteristic properties for the basic ingredient, white wheat flour, were: 14.6% moisture, 12.3% protein, 0.66% ash content, 1.12% fat, and 30.4% wet gluten. The wheat flour also had a gluten deformation index of 3 mm and a falling number index of 356 s. Therefore, the wheat flour used in this study was of a strong quality for bread making and presents low α-amylase activity [32]. Germinated soybean flour (GSF) used as an ingredient in bread presented the following characteristics: 10.5% moisture, 17.9% fat, 40.2% protein, and 5.1% ash content. Germinated soybean flour was characterized by a higher fat and protein content than white wheat flour, mainly due to the fact that the white wheat flour used in this study is refined and it is known that a significant number of nutritive compounds are lost in the refining process. The high difference in fat content is also due to the fact that soybean is a species of legume that it is classified as an oilseed. In a study that we previously carried out, we highlighted the fact that soybean flour without germination had the following characteristics: 9.8% moisture, 16.6% fat, 40.3% protein, and 4.5% ash content [33]. During the germination process, there was a slight increase in the amount of fat from soybean composition, as demonstrated in our previous study. The increase in the amount of fat due to germination can be explained by the fact that during germination, it can be synthesized by some complex lipids, such as phospholipids. Table 1 highlights the chemical characteristics of composite flours with different proportions of GSF supplementation in wheat flour.

### 3.2. Dough Fundamental Rheological Properties

In the case of the creep phase, it was noticed that the data for creep compliance were well adjusted (R^2^ > 0.97) to the Burger’s model. As it may be seen from Figure 1, all rheological parameters were influenced by the addition of GSF into the dough recipe.

Table 2 presents the data determined by performing the creep test and the recovery test, respectively. The J_Co_ parameter, namely the instantaneous compliance of creep phases, decreased significantly (*p* < 0.05) due to supplementation with GSF in wheat flour. The parameter λ_C_, retardation time, increased slightly. The values for creep compliance (J_Co_-instantaneous compliance, J_Cm_-retarded elastic compliance or viscoelastic compliance and J_max_-maximum compliance obtained at the end of test) registered a decreasing trend as the addition level of GSF in wheat flour increased. Furthermore, the µ_Co_ parameter, zero shear viscosity, decreased as the level of GSF addition in wheat flour increased [26].

Regarding the recovery test, it can be observed that the parameters of instantaneous compliance (J_Ro_), retarded elastic compliance evaluated where dough recovery reached equilibrium (J_Rm_), and recovery compliance evaluated where dough recovery reached equilibrium (J_r_) decreased significantly (*p* < 0.05) as the proportion of GSF in wheat flour increased. In the case of the parameter of mean retardation time of recovery phases, λ_R_, the GSF addition into the dough recipe increased significantly (*p* < 0.05). The J_r_/J_max_ ratio registered a slight increase in the case of the supplementation with 5% and 10% GSF and in the case of a GSF addition of 15% and 20%, respectively, and its value was lower compared with the control value.

### 3.3. Dough Microstructure

A variety of microscopic techniques are used for dough microstructure analysis. Among these, a conventional method is light microscopy (i.e., polarizing microscopy, bright field microscopy, fluorescence microscopy). It presents the advantage to observe the changes that occur between dough compounds of a non-deformed sample, allowing for the selective staining of dough compounds. Although the magnification of the light microscopy is modest compared to scanning electron microscopy, it spans the most useful range for food products [34]. Epifluorescence microscopy studies the fluorescence of organic and inorganic compounds simultaneously with absorption and reflection. In general, the excitation of a molecule does not automatically produce fluorescence; therefore, different fluorochromes are used which convert non-fluorescent molecules into fluorescent ones. Fluorochromes are used depending on the affinity they have for special molecular groups, which are detectable with specific spectra under the microscope [35]. In a dough structure, the most used fluorochromes are rhodamine B and fluorescein which stained proteins in red and starch in green. Generally, rhodamine B presents a hydrophobic affinity for protein-rich structures. However, it may also stain starch granules in the protein absence. Nevertheless, the surrounding matrix may affect the visualization of the individual compounds from the dough matrix [36]. By adding GSF to wheat flour, the composition of the dough significantly changed. At an addition level of 20% GSF in the dough recipe, the red area of the dough microstructure significantly increased. This may be explained by the fact that the protein content in the dough structure significantly increased by 45.28%, from 12.3% to 17.87%. The carbohydrate content of the dough decreased by 12.6%, from 71.32 to 62.31%. These carbohydrates include starch, sugars, and fiber. According to Stevensona et al., the starch content of soy is approximately 11% value [37]. Throughout germination, the starch content of soybean seeds decreases as it is hydrolyzed by amylases in order to produce sugars for embryo consumption. Therefore, the germinated soybeans do not present a high amount of starch, which leads to a dough structure with a maximum of 60% starch content when 20% GSF is added to wheat flour.

Figure 2 shows the dough sample microstructures using the EFLM technique. Figure 2A shows the microstructure of the dough without the addition of GSF, and Figure 2B–E show the dough microstructures with GSF incorporated into the dough recipe. As can be seen from the dough microstructure figures, the addition of GSF resulted in an increase in the red colored surfaces. The red color corresponds to rhodamine B which highlights the presence of proteins. Thus, since the size of the red areas increased and their distribution was denser, this translates into the fact that the supplementation with GSF coincided with an augmentation of the protein content in the dough matrix. The green color highlights the presence of starch because the green coloration is specific to fluorescein which bounds to starch granules. It may be seen that the green coloration is more prominent in the control sample.

### 3.4. Bread Quality Evaluation

#### 3.4.1. Bread Physical Characteristics

Table 3 highlights how the physical properties of the bread samples varied due to the addition of GSF. It can be observed that the varying proportions of the GSF addition in bread recipe influenced the bread characteristics differently. Following the 15% GSF addition, there was a significant augmentation (*p* < 0.05) in the specific volume of the bread. At an addition of 20% GSF in the bread recipe, the loaf volume of the bread samples was lower than that of the sample without GSF addition. The same trend was observed in the case of the elasticity and porosity parameters. Thus, it can be noticed that these parameters improved in a significant manner (*p* < 0.05) in the case of the GSF additions of 5%, 10%, and 15%. Conversely, at a 20% GSF supplementation in wheat flour, the porosity and elasticity decreased.

#### 3.4.2. Color Parameters of Bread Samples

Table 4 presents the impact of germinated soybean flour additions on the color parameters of the bread. It is observed that the values of the *L** parameter significantly reduced (*p* < 0.05) in the case of the samples with GSF additions, which indicates that both the bread crumb and crust were darker in the the bread with GSF supplementation. The values of the *a** parameter were significantly higher (*p* < 0.05) for the samples with GSF additions. This shows that the addition of GSF coincided with samples with a redder hue. Furthermore, the values of the parameter *b** increased significantly (*p* < 0.05) as the level of GSF increased. Higher values of the parameter *b** indicate a more intense yellow tint.

#### 3.4.3. Texture Profile Analysis of Bread Samples

Table 5 highlights the fact that supplementation with germinated soybean flour influenced all the determined textural parameters. Thus, the firmness parameter value increased significantly (*p* < 0.05) as the proportion of GSF in the bread recipe increased. At the maximum supplementation of 20% GSF in wheat flour, the value of firmness parameter exceeded twice the value in the control sample. The other textural parameters, gumminess, cohesiveness, and resilience, decreased significantly (*p* < 0.05) due to the addition of GSF.

#### 3.4.4. Crumb Structure of Breads Samples

According to Figure 3, supplementation with GSF had a desirable influence on the crumb microstructure of the bread samples, characterized by smaller pores that are more evenly distributed, in comparison with the sample without GSF addition as seen in Figure 3A. From Table 6, it may be observed that at 10% GSF addition the pore size started to increase slightly but then decreased when 15% and 20% GSF were incorporated into the bread recipe.

Compared with the control sample, the samples with the addition of GSF in wheat flour were characterized by a significantly (*p* < 0.05) lower pore density. Pore circularity did not change much. In contrast, the cell area fraction was influenced differently depending on the percentage of GSF addition in wheat flour.

#### 3.4.5. Sensory Analysis of Bread Samples

According to Figure 4, the supplementation with 5%, 10%, and 15% GSF resulted in a positive trend on the organoleptic properties of the bread samples. The panelists participating in the study appreciated the sample with 15% addition of GSF in wheat flour the most. However, taste and smell were best scored in the case of the sample with 10% GSF supplementation. The addition of 20% GSF worsened the sensory properties.

#### 3.4.6. Effect of GSF Addition on Bread Compositional Analysis

Table 7 highlights that the addition of GSF influenced the nutritional value of the bread samples. From this table, it can be seen that the addition of GSF significantly influenced (*p* < 0.05) the protein, fat, ash, and carbohydrate content of the bread. This may be explained by the fact that germinated soybean flour contains a higher amount of protein, ash, fat, and a lower amount of carbohydrates than white wheat flour.

## 4. Discussion

The values of the J_Co_ and J_Cm_ parameters decreased as the proportion of GSF supplementation in wheat flour increased, which shows that the dough resistance to deformation was improved [38]. The data obtained in the present study are also supported by other studies carried out by different specialists in the field. In these studies, it was concluded that supplementation with roasted chickpea flour coincided with a diminution of the values of the instantaneous and viscoelastic compliance parameters, which indicates a dough with a higher resistance to deformation and flow [39,40]. This shows that dough with legume flour additions may require more energy in order to obtain a deformation. In general, the decrease in dough elasticity is correlated with the diminution of the amount of gluten proteins, in this case due to the addition of GSF in dough recipe [25]. Increasing the value of the parameter λ_C_ (retardation time) indicates that a longer time was required for the viscoelastic deformation of the dough to occur [41]. Struck et al. [42] pointed out that the increase in dough stiffness may be due to the interaction between wheat proteins and dietary fibers. This may be due to the fact that germinated soybean flour has in its composition a higher amount of dietary fiber compared with white wheat flour. The data from the creep test suggests that the addition of GSF led to the modification of the ability of the dough to flow, since lower values of the parameter μ_Co_ were recorded when the percentage of the GSF addition increased [43]. A decrease in the J_max_ parameter values with increasing GSF addition levels indicates that dough deformation occurred more slowly [44,45], i.e., dough resistance to deformation increased.

The instantaneous compliance of recovery phases decreased due to the GSF addition in wheat flour, which means that the elasticity of the dough decreased. This occurs because the amount of gluten in the dough matrix decreased, as it was replaced by the constituents from the GSF flour used as an addition [25]. The decrease in the recovery of the dough due to supplementation with GSF indicates that a certain breakage of elastic bonds occurred [42]. It was also noticed that the retarded elastic recovery of the dough was slowed down as the level of the GSF addition in wheat flour increased. This can be attributed to the higher values of the parameter λ_R_. Increasing the value of the J_r_/J_max_ ratio means a correction of the elastic properties of the dough which was obtained for the samples with 5% and 10% GSF supplementation in wheat flour. In contrast, at 15% and 20% GSF additions, the value of the ratio was lower than that of the control. This indicates a decrease in dough elasticity.

According to the EFLM images, the dough sample from Figure 2A (the sample without GSF addition) was highlighted by the largest areas colored in green. This indicates the fact that the control sample had the highest starch content. In contrast, it can be seen from Figure 2B–E that the areas colored in red increased progressively with the increase in GSF addition. This indicates that the starch from the wheat flour was replaced by a larger amount of protein from the germinated soybean flour content. This was predictable since it was already reported that white wheat flour has a lower quantity of protein than soybeans [46], with an amount four times higher than that found in wheat flour [47]. From the dough microstructure it can also be seen that, in the case of the sample without GSF addition and those with low levels of GSF supplementation in wheat flour, the starch granules appear to be dispersed and concentrated in several areas. Furthermore, as the level of GSF addition increased, the starch granules were dispersed among proteins, being surrounded and isolated by them. Moreover, it may be seen that no black areas appear in the dough system, which means that dough matrix is homogeneous and that the GSF additions did not cause a weakening of the gluten network even when the proportion of the addition was high.

The positive influence on the specific loaf volume of the bread samples due to the GSF addition in wheat flour may be due to the fact that while germination takes place, there is an increase in fermentable sugars due to the action of amylases on starch [48]. Dextrins and reducing sugars obtained through starch hydrolyses by amylases [49] during the germination process play a role in plumule development [50] and in the bread making process in the improvement of the baker’s yeast activity in the fermentation stage. The result is the release of a larger amount of CO_2_ and, implicitly, a higher volume of the bread. The data from this study are in agreement with other studies in the field. For example, Heberle et al. [51] highlighted the fact that bread samples with sprouted rice flour were characterized by a larger volume than those obtained with unsprouted rice. Shin et al. [52] concluded that the bread obtained from germinated soybeans had a higher specific volume than the bread obtained from non-germinated soybeans. Furthermore, different studies concluded that some modifications take place during sprouting which influence the starch gelatinization and modify the aggregation of proteins from the composition of grains that were germinated. Thanks to germination, the ability of leguminous flours to foam and emulsify is improved [53]. The improvement in the proteolytic activity during germination leads to the degradation of the soybean’s storage proteins, which slightly decreased [33]. The final result is the increased level of free amino acids and shorter peptides, which lead to the improvement of protein solubility. The foaming capacity is related to the volume of air that the protein from the soybean composition can incorporate into the dough system [54]. Different studies have emphasized that, in the case of chickpea and hemp seeds, germination leads to the obtaining of protein isolates with a higher emulsifying activity [55]. At an addition level of 20% germinated soybean flour, the specific volume of the bread samples was significantly lower (*p* < 0.05) than those for the samples without GSF addition, which means that a supplementation with high proportions of GSF resulted in a decrease in the dough capacity in retaining CO_2_ in the dough network because of the weakening process of the gluten matrix [56].

Regarding the porosity of the bread crumb, supplementation with GSF caused a significant improvement in bread porosity (*p* < 0.05) up to a proportion of 15% GSF supplementation in wheat flour. At 20% GSF addition, the porosity was significantly lower (*p* < 0.05) in comparison with the control without GSF addition. The influence of the bread porosity due to supplementation with GSF is explained by the fact that it increased the capacity of the dough network to produce CO_2_, but also to retain it in the dough system. According to our previous study, GSF addition in dough recipes leads to a significant decrease in the falling number value [21] and the α-amylase activity, which will act on starch to increase the number of fermentable sugars resulting in an increase in carbon dioxide production by yeasts. This produces a more porous crumb. The present study is consistent with other studies in the field. Cardone et al. [57] highlighted the possibility of using germination to obtain bread with a better porosity. Marti et al. [58] demonstrated that the addition of 50% wheat flour germinated for 24 h resulted in bread with a better porosity.

According to our data, crumb elasticity improved in a significant manner (*p* < 0.05) following the addition of 5%, 10%, and 15% germinated soybean flour in the bread recipe. The improvement in the elasticity of the crumb may be due to the α-amylase enzyme, whose activation occurs due to germination and has a positive effect on this bread quality parameter. High crumb elasticity was associated by consumers with high quality bread. The fact that enzymes generally improve bread elasticity has also been discussed in numerous other studies [59,60]. According to them, amylases lead to the production of fermentable sugars which increases the dough capacity for retaining CO_2_ in the dough system. Furthermore, other enzymes, such as xylanases, influence the elasticity of the gluten network and improve the quality of bread [61].

The significant decrease (*p* < 0.05) in the *L** parameter for the crust and the crumb produced darker bread samples as the GSF proportion of addition in wheat flour increased. This occurs due to the increase in the amount of reducing sugars and amino acids due to germination, compounds that intensify the Maillard reactions that occur during the bread baking stage [62]. The ways in which the increase in the amount of reducing sugars and the number of amino acids occurs have been explained in the specialized literature. Wang et al. (2020) concluded that higher enzyme activity coincides with an intense erosion of starch granules and with a decrease in relative crystallinity and short-range ordered degrees, and the unwinding of the double helix structure [45]. Amylases are enzymes that catalyze the hydrolysis of starch. The purpose of this hydrolysis is to obtain oligosaccharides, i.e., dextrin and maltose. An intensification of the Maillard reactions means a specific browning of the bread samples, which leads to bread with a crust and crumb that are darker. Millar et al. [63] reported similar results, namely that the bread samples darkened in color due to the addition of pea flour in the bread recipe. Thus, in the case of samples with GSF supplementation, there was a significant decrease (*p* < 0.01) in the *L** (brightness) index, both in the case of the crust and the bread crumb. Millar et al. [64] showed that the incorporation of pea flour leads to the intensification of browning reactions due to the augmentation of the quantity of proteins from the dough. Thus, it can be concluded that in the present study, the Maillard reactions were favored by a higher protein content due to the fact that germinated soybean flour has a higher protein quantity than white wheat flour.

The Maillard reactions were intensified in the bread containing the GSF addition in wheat flour because it has a higher amount of maltose resulting from more intense activity of the amylase that hydrolyzed the starch [65]. In the last stages of the Maillard reactions, specific browning compounds, such as melanoidins, are formed in the bread crust, which are heterogeneous polymers. Melanoidins from the bread crust had a protein skeleton. They consisted of specific color compounds in association with polymer gluten, as pointed out by Borelli et al. [66] in their study. The fact that the browning reactions were enhanced in the samples with GSF addition indicates a higher amount of melanoidins, which is desirable for several reasons. Different studies have highlighted the positive aspects related to melanoidins, such as anti-inflammatory [67] and antioxidant [68] properties, bifidogenic effects [66], antimicrobial properties, and positive effects in reducing the multiplication of cancer cells in the colon and gastric mucosa [69]. Another reason for the decrease in brightness is that the GSF contributed to the increase in the amount of protein in the bread content. Different studies have already concluded that a higher amount of protein leads to a darker bread color [70].

The darkening of bread samples was also obtained when ginger powders were added to the bread recipe [71]. Xu et al. [72] correlated the darkening of bread samples with a higher content of phenolic compounds due to the addition of black tea or green tea powder. The germination contributes to the augmentation of the number of phenolic compounds [73] and, therefore, to the change in bread color following GSF additions.

The incorporation of GSF in white wheat bread recipes coincided with a significant increase (*p* < 0.05) in the *a** parameter for the bread crumb and also for the bread crust. This indicates that the GSF incorporation resulted in an increase in the red tint of the bread samples. The intensification of the red coloration can be correlated with the Maillard and browning reactions that took place during the baking stage. At the same time, the GSF addition led to bread samples with significantly higher values (*p* < 0.05) of the *b** parameter, which indicates an intensification of yellow coloration. The explanation behind this may be the presence of yellow pigments from the soybean compounds [74].

The growth in a significant manner (*p* < 0.05) in the firmness parameter of the bread as the proportion of the GSF incorporation in wheat flour increased can be assigned to the high amount of soluble dietary fibers in GSF [75,76]. Therefore, with the increased proportion of GSF in the dough matrix, the interaction between dough compounds changed. This leads to a diminution in the gluten network strength, which may be attributed to a higher ability of soluble fibers from GSF to bind to water which exists in the dough system. At the same time, thanks to the GSF addition, there was a gluten dilution effect in the dough network, which may lead to an increase in the value of the firmness parameter. The results of the present study are supported by studies that have been conducted previously. In these studies, different types of fibers from wheat and oats had a similar effect on bread [77]. The GSF incorporation in the white wheat bread recipe resulted in an increase in the number of proteins, which modify their interaction with the starch from the dough network. The gumminess of the bread samples decreased with the increase in the proportion of GSF additions into the dough system due to the changes in the gluten network. The GSF addition led to a dilution of the gluten in the dough network, which also coincided with a decrease in the cohesiveness and resilience parameters. Different studies have highlighted the fact that lower bread cohesiveness may be due to a higher number of fibers in the bread recipe which may cause a decrease in the interaction between the dough components and water [78]. The diminution of the cohesiveness parameter was also reported in another study when bran was added to wheat flour [79]. A decrease in the resilience parameter can be attributed to a gluten network weakening phenomenon which occurs due to the large number of insoluble fibers from the GSF content, as explained in previous studies [80].

The positive effect of supplementation with GSF in the white wheat bread recipe on the structure of the bread crumb can be attributed to the more intense activity of enzymes specific to the dough system of GSF incorporated in wheat flour. During the germination process, the enzyme activity of soybeans increased. In a previous study, we highlighted how germination resulted in increased amylase activity. Thus, we highlighted the fact that germination led to a decrease in the value of the falling number index [21]. This test indicates amylase activity in wheat flour. Enzymes have an imperative role in increasing the number of fermentable sugars [81] due to the starch hydrolysis process. The fermentable sugars are successfully used by the yeast. This results in an augmentation in the amount of CO_2_ produced, which should have led to smaller pores in the bread structure. The addition of GSF caused a decrease in the dough elasticity and an increase in the resistance to deformation [82]. The fact that the pore size decreased can be attributed to the fact that the gluten network was less able to retain gases in the dough network because the amount of gluten decreased with the addition of GSF to the dough recipe.

The appearance of the bread was positively marked by GSF incorporation due to several considerations: improving the bread loaf volume, improving the porosity and elasticity, and favoring the browning reactions during the baking stage. The control sample was obtained only from wheat flour with no additives. It was of a strong quality for bread making and had a low α-amylase activity. This sample presented low values for specific volume, porosity, and elasticity compared with the samples with 5–15% GSF additions, values which affected the sensory evaluation given by the panelists. The color of the bread had an improved score up to a maximum of 15% GSF addition, mainly due to the enhancement from the Maillard reactions which produced bread samples with a darker crust. The bread crumb had a slightly more yellowish coloration due to specific pigments in soybeans. The taste of the samples with the GSF addition was more appreciated. This can be attributed to the amylase enzyme activation after germination of soybeans has taken place. This may have an important role in imprinting a certain sweetness to the bread. Amylolytic enzymes are used in the bakery industry to act on the starch and to produce reducing sugars that will be used by the yeast. Thus, the addition of GSF imparts a slight sweetness to the bread samples thanks to the enzyme activation that takes place following the germination process of soybeans [83]. Sweetness is enjoyed by consumers, so it is understandable that the breads with the GSF additions were more appreciated by the panelists in terms of taste indicators. The texture of the bread samples was positively appreciated by the consumers in the case of the samples with a maximum of 15% GSF incorporation, mainly thanks to the enzyme activation that takes place following the start of the germination process with enzymes that acted positively in the dough system. The flavor of the samples with the incorporation of GSF was more appreciated, probably due to volatile compounds formed during baking that were intensified due to a higher protein content in the samples with GSF additions [63]. During the Maillard reactions, α-amino acids were converted into aldehydes [84] and other volatile flavor compounds. An improvement in the organoleptic properties of the bread samples was also demonstrated by Rizzello et al., who concluded that an incorporation of a maximum of 15% leguminous flours, such as chickpea, lentil, or bean flour, in wheat flour led to bread samples which were better scored by consumers, referring to sensory attributes such as color, flavor, and taste [85].

The nutritional value of the bread samples was significantly influenced (*p* < 0.05) by the addition of GSF in wheat flour. At a maximum level of 20% GSF addition in wheat flour, the proteins increased by 3.85% as the germinated soybean flour contains 27.9% more proteins than white wheat flour. Furthermore, the amount of fat increased significantly (*p* < 0.05) due to the GSF addition, i.e., by 2.52%, in the case of the maximum level of GSF incorporated in the bread recipe. This is explainable as germinated soybean flour contains 16.78% more fat. The amount of ash increased significantly (*p* < 0.05), being 0.55% higher in bread samples with 20% GSF. Conversely, the amount of carbohydrates decreased significantly (*p* < 0.05) by 7.01% in bread samples with 20% GSF. This may be due to the fact that the amount of refined wheat flour with a high starch content used in bread recipes decreased, and the amount of germinated soybean flour increased, which contains a lower amount of carbohydrates. Regarding the increase in the amount of protein, fat, and ash and the decrease in the amount of carbohydrates, similar results were recorded in a study that we carried out previously, in which we analyzed the influence of adding germinated bean flour to wheat flour on the bread quality [86].

## 5. Conclusions

GSF incorporation in wheat flour had a positive impact on the dough rheology and on bread quality properties. In terms of rheology, the addition of germinated soybean flour coincided with a decrease in the value of the parameters of instantaneous compliance of creep phases, retarded elastic compliance of creep phases, zero shear viscosity, maximum creep compliance obtained at the end of the creep test, instantaneous compliance of recovery phases, and retarded elastic compliance or viscoelastic compliance of recovery phases. This indicates the fact that the GSF addition improved the ability of the dough to resist to deformation. Furthermore, the elastic properties of the dough were improved when low proportions of GSF were added to the dough and decreased when 15% and 20% GSF were added to wheat flour. The dough microstructure obtained by EFLM indicated an increase in the amount of protein due to GSF addition. The specific volume, porosity, and elasticity of the bread with a maximum of 15% GSF incorporation were improved. The bread samples with GSF incorporation recorded lower values of the parameter *L** (brightness) and higher values for the parameters *a** (shade of red) and *b** (shade of yellow), both from the point of view of the crust and bread crumb. The bread with the GSF addition was highlighted by a more pronounced firmness but had a lower gumminess, cohesiveness, and resilience values. The addition of GSF led to bread samples with a crumb structure characterized by smaller pores that were more evenly distributed. The panelists appreciated the bread sample with 15% GSF addition the most. The bread compositional values indicate a significant increase in protein, fat, and ash by GSF addition to the bread recipe. Taking into account the data presented above, it can be deduced that germinated soybean flour can be used as an incorporation in bread recipes to produce bakery products with an improved quality.

## Figures and Tables

**Figure 1 foods-12-01316-f001:**
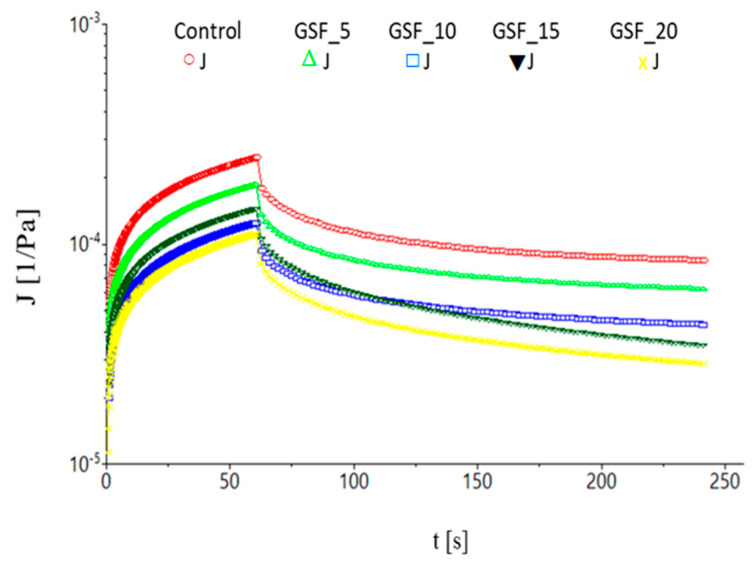
Creep and recovery curves of wheat flour dough with different levels (-o-0%; -∆-5%; -□-10%; -▼-15%; -x-20%) of germinated soybean flour additions.

**Figure 2 foods-12-01316-f002:**
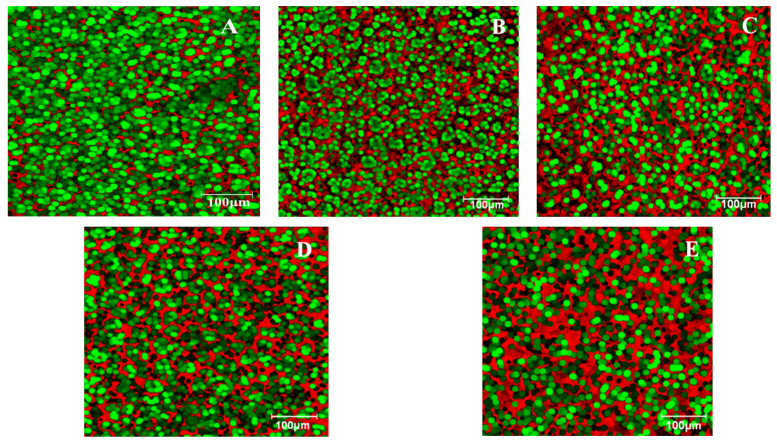
Microstructure measured by EFLM of the wheat dough with GSF in varying proportions: 0% (**A**), 5% (**B**), 10% (**C**), 15% (**D**), and 20% (**E**). Red, protein; green, starch granules.

**Figure 3 foods-12-01316-f003:**
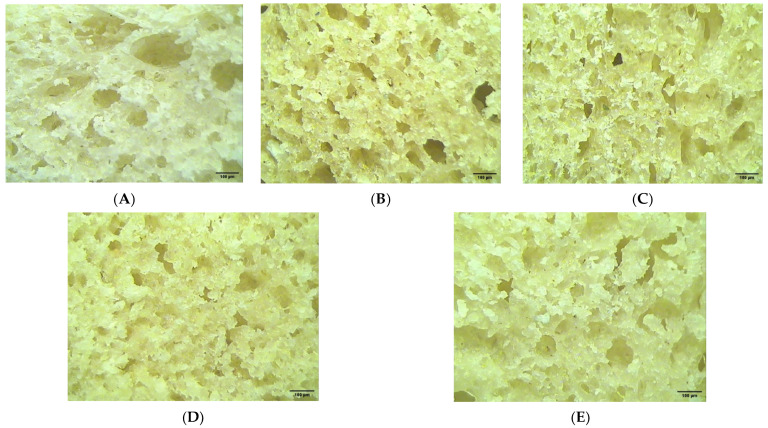
Structure of wheat dough with germinated soybean flour (GSF) at different levels: 0% (**A**), 5% (**B**), 10% (**C**), 15% (**D**), and 20% (**E**).

**Figure 4 foods-12-01316-f004:**
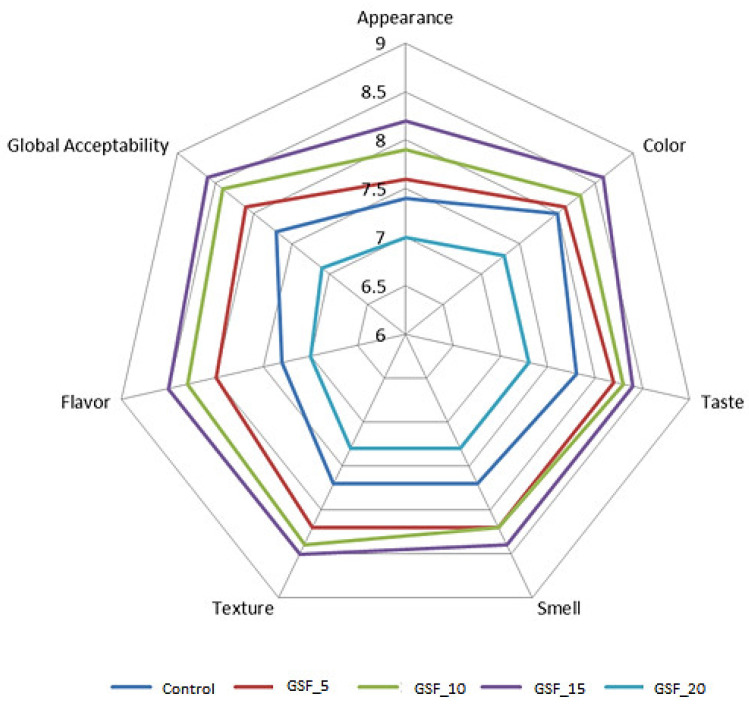
Sensory analysis for bread samples.

**Table 1 foods-12-01316-t001:** Chemical characteristics of composite flours with different proportions of GSF supplementation [33].

Sample	Protein (%)	Fat (%)	Ash (%)	Moisture (%)	Carbohydrate (%)
Control	12.30 ± 0.01 ^e^	1.12 ± 0.00 ^e^	0.65 ± 0.00 ^e^	14.60 ± 0.01 ^a^	71.32 ± 0.01 ^a^
GSF_5	13.69 ± 0.00 ^d^	1.96 ± 0.00 ^d^	0.87 ± 0.01 ^d^	14.39 ± 0.01 ^b^	69.08 ± 0.00 ^b^
GSF_10	15.09 ± 0.01 ^c^	2.79 ± 0.01 ^c^	1.08 ± 0.02 ^c^	14.18 ± 0.01 ^c^	66.82 ± 0.01 ^c^
GSF_15	16.48 ± 0.01 ^b^	3.63 ± 0.01 ^b^	1.30 ± 0.03 ^b^	13.97 ± 0.02 ^d^	64.57 ± 0.01 ^d^
GSF_20	17.87 ± 0.01 ^a^	4.47 ± 0.01 ^a^	1.51 ± 0.04 ^a^	13.76 ± 0.03 ^f^	62.31 ± 0.01 ^e^

The results are mean ± standard deviation (*n* = 3). Composite flours containing germinated soybean flour, GSF: a–f, mean values followed by the same letter within a column were not significantly different (*p* < 0.05).

**Table 2 foods-12-01316-t002:** Parameters of Burger′s model.

Samples	Creep Phase					Recovery Phase				
	J_Co_ × 10^5^ (Pa^−1^)	J_Cm_ × 10^5^ (Pa^−1^)	λ_C_ (s)	μ_Co_ × 10^−6^ (Pa·s)	J_max_ × 10^5^ (Pa^−1^)	J_Ro_ × 10^5^ (Pa^−1^)	J_Rm_ × 10^5^ (Pa^−1^)	λ_R_ (s)	J_r_ × 10^5^ (Pa^−1^)	J_r_/J_max_ (%)
Control	6.93 ^e^ (0.02)	20.00 ^c^ (0.01)	34.99 ^a^ (0.05) ^ab^	0.57 ^c^ (0.00)	24.76 ^e^ (0.02)	8.66 ^e^ (0.04)	8.32 ^e^ (0.00)	34.36 ^a^ (0.00)	16.98 ^e^ (0.04)	68.57 ^b^ (0.14)
GSF_5	5.18 ^d^ (0.01)	20.00 ^c^ (0.00)	35.60 ^ab^ (0.08)	0.12 ^a^ (0.01)	18.64 ^d^ (0.02)	6.47 ^d^ (0.02)	6.32 ^d^ (0.00)	34.84 ^b^ (0.00)	12.79 ^d^ (0.02)	68.61 ^b^ (0.03)
GSF_10	3.33 ^b^ (0.03)	10.01 ^b^ (0.01)	35.97 ^bc^ (0.00)	0.13 ^a^ (0.02)	12.51 ^b^ (0.02)	4.47 ^c^ (0.03)	4.30 ^a^ (0.00)	36.63 ^c^ (0.00)	8.77 ^b^ (0.03)	70.08 ^c^ (0.09)
GSF_15	3.98 ^c^ (0.07)	10.00 ^b^ (0.00)	36.50 ^cd^ (0.58)	0.18 ^b^ (0.00)	14.52 ^c^ (0.02)	3.62 ^b^ (0.02)	6.13 ^c^ (0.00)	44.84 ^e^ (0.00)	9.75 ^c^ (0.02)	67.13 ^a^ (0.02)
GSF_20	3.02 ^a^ (0.03)	9.73 ^a^ (0.21)	36.94 ^c^ (0.08)	0.16 ^b^ (0.01)	11.15 ^a^ (0.02)	3.01 ^a^ (0.02)	4.49 ^b^ (0.00)	42.55 ^d^ (0.00)	7.50 ^a^ (0.02)	67.27 ^a^ (0.06)

Values in parentheses are standard deviations. Means followed by the same letter within a column are not significantly different. Different letters (^a,b,c,d,e^) within the same column for each parameter indicate that means are significantly different (*p* < 0.05).

**Table 3 foods-12-01316-t003:** Physical characteristics of the bread with different proportions of GSF supplementation in wheat flour.

Bread Samples	Specific Volume (cm^3^/100 g)	Porosity (%)	Elasticity (%)
Control	331.5 ± 0.74 ^b^	67.4 ± 0.86 ^b^	91.3 ± 0.57 ^b^
GSF_5	339.1 ± 0.73 ^c^	70.6 ± 0.47 ^c^	91.9 ± 0.24 ^bc^
GSF_10	349.1 ± 0.82 ^d^	72.3 ± 0.30 ^d^	93.0 ± 0.06 ^cd^
GSF_15	357.2 ± 1.49 ^e^	73.7 ± 0.46 ^d^	94.4 ± 0.44 ^d^
GSF_20	327.6 ± 0.55 ^a^	65.4 ± 0.50 ^a^	88.6 ± 0.81 ^a^

The results are mean ± standard deviation (*n* = 3). Bread samples containing germinated soybean flour, GSF: a–e, mean values followed by the same letter within a column were not significantly different (*p* < 0.05).

**Table 4 foods-12-01316-t004:** Color parameters of the bread with different proportions of GSF supplementation in wheat flour.

Bread Samples	Crust Color			Crumb Color		
	L*	a*	b*	L	a*	b*
Control	76.25 ± 0.94 ^d^	3.44 ± 0.27 ^a^	3.14 ± 0.43 ^a^	66.37 ± 0.88 ^d^	−4.62 ± 0.32 ^e^	1.69 ± 0.22 ^a^
GSF_5	75.55 ± 0.43 ^d^	4.93 ± 0.07 ^b^	8.96 ± 0.14 ^b^	64.11 ± 0.12 ^b^	−3.63 ± 0.27 ^d^	4.59 ± 0.31 ^b^
GSF_10	71.74 ± 0.98 ^c^	6.36 ± 0.33 ^c^	10.36 ± 0.44 ^b^	61.03 ± 0.59 ^b^	−2.86 ± 0.14 ^c^	6.93 ± 0.08 ^c^
GSF_15	68.27 ± 0.84 ^b^	9.37 ± 0.22 ^d^	12.18 ± 0.20 ^c^	57.63 ± 0.41 ^a^	−1.80 ± 0.14 ^b^	9.14 ± 0.15 ^d^
GSF_20	59.46 ± 0.50 ^a^	13.33 ± 0.40 ^e^	15.93 ± 0.78 ^d^	57.21 ± 0.31 ^a^	−0.78 ± 0.15 ^a^	11.04 ± 0.15 ^e^

The results are mean ± standard deviation (*n* = 10). Bread samples containing germinated soybean flour, GSF: a–e, mean values followed by the same letter within a column were not significantly different (*p* < 0.05).

**Table 5 foods-12-01316-t005:** Texture parameters of the bread with different proportions of GSF supplementation in wheat flour.

Bread Samples	Firmness (*N*)	Gumminess (*N*)	Cohesiveness (Adimensional)	Resilience (Adimensional)
Control	9.01 ± 3.06 ^a^	7.23 ± 1.73 ^b^	0.82 ± 0.03 ^c^	1.72 ± 0.04 ^d^
GSF_5	12.78 ± 0.48 ^ab^	6.10 ± 0.72 ^ab^	0.50 ± 0.02 ^a^	1.67 ± 0.02 ^d^
GSF_10	16.38 ± 0.68 ^bc^	5.56 ± 0.42 ^ab^	0.70 ± 0.02 ^b^	1.58 ± 0.03 ^c^
GSF_15	18.70 ± 0.28 ^c^	4.37 ± 0.54 ^a^	0.57 ± 0.02 ^a^	1.42 ± 0.02 ^b^
GSF_20	20.34 ± 1.04 ^c^	6.70 ± 0.30 ^ab^	0.54 ± 0.03 ^a^	1.30 ± 0.01 ^a^

The results are mean ± standard deviation (*n* = 3). Bread samples containing germinated soybean flour, GSF: a–d, mean values followed by the same letter within a column were not significantly different (*p* < 0.05).

**Table 6 foods-12-01316-t006:** Computed bread crumb characteristics.

Sample	Pores Density (1/cm^2^)	Mean Cell Size (mm^2^)	Pore Circularity (Adimensional)	Cell Area Fraction (%)
Control	14.13 ± 0.82 ^e^	0.14 ± 0.02 ^c^	0.90 ± 0.08 ^a^	0.68 ± 0.09 ^a^
GSF_5	10.65 ± 0.64 ^b^	0.10 ± 0.01 ^ab^	0.89 ± 0.07 ^a^	0.76 ± 0.08 ^b^
GSF_10	12.99 ± 0.78 ^d^	0.12 ± 0.02 ^bc^	0.88 ± 0.03 ^a^	1.49 ± 0.15 ^d^
GSF_15	9.24 ± 0.59 ^a^	0.09 ± 0.01 ^a^	0.89 ± 0.05 ^a^	0.68 ± 0.07 ^a^
GSF_20	11.61 ± 0.64 ^c^	0.11± 0.01 ^abc^	0.89 ± 0.09 ^a^	1.11 ± 0.18 ^c^

The results are mean ± standard deviation (*n* = 3). Bread samples containing germinated soybean flour, GSF: a–e, mean values followed by the same letter within a column were not significantly different (*p* < 0.05).

**Table 7 foods-12-01316-t007:** Compositional analysis of bread samples with different levels of germinated soybean flour (SGF).

Bread Samples	Moisture (%)	Protein (%)	Fat (%)	Ash (%)	Carbohydrates (%)	Energy (kcal/100 g)
Control	44.72 ± 0.02 ^ab^	8.80 ± 0.01 ^a^	0.81 ± 0.01 ^a^	0.51 ± 0.01 ^a^	45.14 ± 0.04 ^e^	223.13 ± 0.14 ^a^
SGF_5	44.81 ± 0.01 ^c^	9.73 ± 0.02 ^b^	1.38 ± 0.01 ^b^	0.61 ± 0.01 ^b^	43.46 ± 0.01 ^d^	225.18 ± 0.03 ^b^
SGF_10	44.75 ± 0.02 ^b^	10.81 ± 0.01 ^c^	2.04 ± 0.02 ^c^	0.76 ± 0.01 ^c^	41.64 ± 0.01 ^c^	228.16 ± 0.04 ^c^
SGF_15	44.69 ± 0.03 ^a^	11.64 ± 0.03 ^d^	2.62 ± 0.02 ^d^	0.92 ± 0.01 ^d^	40.12 ± 0.03 ^b^	230.66 ± 0.09 ^d^
SGF_20	44.83 ± 0.03 ^c^	12.65 ± 0.03 ^e^	3.33 ± 0.02 ^e^	1.06 ± 0.01 ^e^	38.13 ± 0.05 ^a^	233.09 ± 0.06 ^e^

The results are mean ± standard deviation (*n* = 3). Means followed by the same letter within a column are not significantly different (*p* < 0.05).

## Data Availability

The datasets generated for this study are available on request to the corresponding author.

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
