# Peer review of "Dough Rheological Behavior and Bread Quality as Affected by Addition of Soybean Flour in a Germinated Form"

_foods, 2023, doi:10.3390/foods12061316_

Round 1

Reviewer 1 Report

Materials and methods. Author must add that sensory analysis was approved by a bioethical committee.

-          I think that author must include soybean flour without germination in experiments.  Title has the word “germinated” , so experimentation must include soybean without germination to prove that germination is the factor that change the quality.

 Author must include the compositional analysis of the doughs with different levels of germinated bean flour (GBF) additions.

Results:

Line 186-192. Explain in text why for GSF increased up 17.9% and 40.2% fat and protein respectively.

Line 191. Add  the characteristics for soybean flour without germination. Discuss in text.

Line 221.  “microsturctures”.

Line 219. 3.3. Dough microstructure.   Comment: This technique is optical and does not measure microstructure. Discuss in text why this technique is considered a microstructural test.

Fig 1. According with image the red color is protein (rhodamine) . Why if you increase 20% of GSF, there are as 50% of increases in red color (Fig 1.e) . Discuss in text.

Note: Rhodamine is a marker colorant that is not approved by FDA.

Line 290. Fig 3. Why control has as poor values? I think that control must have the highest values and the mixes reaching those values. Isnt it?. Discuss in text

Line 282. I can not see that GSF addition increase pore size. I see in fig 2 that pore  size reduce as GSF increase.  So , I suggest to author that measure the size pore for each plot of figure 2 and add a table showing size and discuss it.

Line 486-489 . Author must prove the improved quality of the mixes as Rizzello et al proved it. How measure quality bread? Include in materials and method how to measure the quality bread. So, author can include in conclusion that quality bread was improved.

Conclusion: In general, mixes increasing the firmness and this means that mixes flour were more hard and I think that this parameter not improve the quality bread. Why author conclude that the decrease of firmness is better for the quality bread?

References: Add in reference list this article: Foods 2021, 10, 1542. https://doi.org/10.3390/foods10071542.

Author Response

16 March 2023

Dear Referee,  

We would like to thank the referee for the close reading and for the proper suggestions. We hope that we provide all the answers to the reviewer’s comments.

Thank you very much for the recommendations to publish our paper entitled “Dough Rheological Behavior and Bread Quality as Affected by Addition of Germinated Soybean Flour”.

The present version of the paper has been revised according to the reviewer’s suggestions.             

We uploaded the corrected version of the article for which we used the red color for the addition text.

GENERAL COMMENTS:

Referee comments: Materials and methods. Author must add that sensory analysis was approved by a bioethical committee.

Response: We would like to thank to the referee for his/her suggestions. We specified now in the materials and methods section, Lines 197-201, that the sensory analysis methodology was approved by the ethics committee of our faculty department and that the participants gave their consent to participate.

Referee comments:  Author must include the compositional analysis of the doughs with different levels of germinated bean flour (GBF) additions.

Response: We would like to thank to the referee for his/her suggestions. We completed now the manuscript with the chemical characteristics of composite flours with different proportions of GSF supplementation (see table 1) – Lines 232-242.

Referee comments:  I think that author must include soybean flour without germination in experiments.  Title has the word “germinated”, so experimentation must include soybean without germination to prove that germination is the factor that change the quality.

Response: We want to thank to the referee for his/her suggestions. Indeed was a confusion due to the title used for our manuscript. We changed now as Dough Rheological Behavior and Bread Quality as Affected by Addition of Soybean Flour in a Germinated Form – line 3. More, we completed now our manuscript with data of soybean characteristics which was not germinated (fat, protein and ash value). These data can be viewed in Table 1 – Lines 232-242. As may be seen, according to the data obtained, are significant differences between germinated and non-germinated soybean values. Until now, the use of germinated seed flour in bread making was widely used on an industrial level by using malt from barley (raw material in beer technology) produced by a gentle drying in order to preserve it enzymatic activity. The idea of our research was to identify and further develop other ingredients (in our case germinated soybean flour) which avoid or at least reduce the use of replacers like additives in wheat flour by achieving natural and good bakery products from the nutritional and technological point of view. In our study the wheat flour does not contain additives and is with a low alpha amylase activity (has a high falling number values). Throughout germination the soybean enzymatic activity increased (we reported the falling number value descreased in other previous research) which we mentioned it in our manuscript, in Lines 487-491 (see: https://www.mdpi.com/2076-3417/11/24/11706. We did some experiments and with no germinations the falling number of mix flours between soy and wheat does not decreased very much. So, the germinated soybean flour is a completly different ingredient that the native soybean one which besides an ingredient that improves bread nutritional valus it also improves it technological characteristics due to it enzymatic activity (of course for wheat flours  with a high falling number values).

Referee comments: Line 186-192. Explain in text why for GSF increased up 17.9% and 40.2% fat and protein respectively.

Response: We want to thank to the referee for the close reading of our manuscript. The 17.9% and 40.2% are the values for germinated soybean flour for fat and protein. Wheat flour is a refined one which present a much lower protein and fat content. We explained in a more extensive way now why is the difference, in Lines 214-230. I think was a confusion due to the English language (it is not an increased are the GSF data). We have reviewed the formulation.

Referee comments: Line 191. Add  the characteristics for soybean flour without germination. Discuss in text.

Response: We added the characteristics for soybean flour without germination (see Table 1 - – Lines 232-242) and explained the reasons behind the changes that occur in the characteristics of soybean flour due to germination, Lines 214-230.

Referee comments: Line 221.  “microsturctures”.

Response: We have modified the form of the word in the manuscript, as can be seen in Line 301. Thank you very much for pointing us this mistake.

Referee comments: Line 219. 3.3. Dough microstructure.   Comment: This technique is optical and does not measure microstructure. Discuss in text why this technique is considered a microstructural test.

Response: We completed and discussed in our manuscript that EFLM is an optical one and why is prefered for microstructural analysis (see Lines 272-288). We have reported in particular to the article published by – see reference 35 :

https://www.sciencedirect.com/science/article/abs/pii/S0733521009001386

Referee comments:Fig 1. According with image the red color is protein (rhodamine) . Why if you increase 20% of GSF, there are as 50% of increases in red color (Fig 1.e) . Discuss in text.

Response: We have now discussed in the manuscript (Lines 288-298) in a more extensive way why this changes occured.  

Referee comments: Note: Rhodamine is a marker colorant that is not approved by FDA.

Response: We agree with the referee point of view. In the research studies it is the most used fluorochromes for dough microstructure. We specified this in Lines 284-285.

Referee comments: Line 290. Fig 3. Why control has as poor values? I think that control must have the highest values and the mixes reaching those values. Isnt it?. Discuss in text

Response: We believe that this was because that we used a wheat flour with no additivation which was a not very good one for bread making (low alpha amylase activity), of strong quality for bread making (a gluten deformation index of 3 mm). So, using GSF (which has an enzymatic activity) the gluten extension was improved and also the amylase activity and carbon dioxide formed. More, GSF brings some lipids which positively affect bread quality. As it may be seen the control sample has low values for loaf volume, porosity, elasticity compared to the samples with GSF addition which was also reflected in bread sensory quality. We explained in the discussion section (Lines 451-507) the reasons that led to the better appreciation of the samples with 5%, 10% and 15% addition of GSF than the control sample by the participants of the sensory session.

Referee comments: Line 282. I cannot see that GSF addition increase pore size. I see in fig 2 that pore  size reduce as GSF increase.  So , I suggest to author that measure the size pore for each plot of figure 2 and add a table showing size and discuss it.

Response: We want to thank to the referee for his/her suggetions. We completed our manuscript with table 6 – computed bread crumb characteristics which we discussed in the manuscript, in Lines 372-375 and Lines 597-602.

Referee comments: Line 486-489 . Author must prove the improved quality of the mixes as Rizzello et al proved it. How measure quality bread? Include in materials and method how to measure the quality bread. So, author can include in conclusion that quality bread was improved.

Response: We want to thank to the referee for his/her suggetions. We completed our manuscript with table 7 Compositional analysis of bread samples with different levels of germinated soybean flour (SGF) addition which we discussed in the manuscript, Lines 386-390 and Lines 636-652.

Referee comments: Conclusion: In general, mixes increasing the firmness and this means that mixes flour were more hard and I think that this parameter not improve the quality bread. Why author conclude that the decrease of firmness is better for the quality bread?

Response: We agree with the referee point of view. The firmeness parameter does not indicates an improvement of bread quality. However, the other characteristics such as specific volume, porosity, elasticity, sensorial ones, compositional characteristics of bread samples, e.g. make us to conclude that the addition of GSF had an overall effect of improving the bread quality.

Referee comments: References: Add in reference list this article: Foods 2021, 10, 1542. https://doi.org/10.3390/foods10071542.

Response:We have added the recommended article in the reference list, Lines 906-907.

Finally, the authors would like to thank to the reviewer for his/her appreciations and for all the suggestions because these helped us to correct our paper and to optimize it.

Sincerely,

Georgiana Codină et al.

GENERAL COMMENTS:

Referee comments: Materials and methods. Author must add that sensory analysis was approved by a bioethical committee.

Response: We would like to thank to the referee for his/her suggestions. We specified now in the materials and methods section that the sensory analysis methodology was approved by the ethics committee of our faculty department and that the participants gave their consent to participate.

Referee comments:  Author must include the compositional analysis of the doughs with different levels of germinated bean flour (GBF) additions.

Response: We would like to thank to the referee for his/her suggestions. We completed now the manuscript with the chemical characteristics of composite flours with different proportions of GSF supplementation (see table 1).

Referee comments:  I think that author must include soybean flour without germination in experiments.  Title has the word “germinated”, so experimentation must include soybean without germination to prove that germination is the factor that change the quality.

Response: We want to thank to the referee for his/her suggestions. Indeed was a confusion due to the title used for our manuscript. We changed now as Dough Rheological Behavior and Bread Quality as Affected by Addition of Soybean Flour in a Germinated Form. More, we completed now our manuscript with data of soybean characteristics which was not germinated (fat, protein and ash value). As may be seen according to the data obtained are significant differences between germinated and non-germinated soybean values. Until now, the use of germinated seed flour in bread making was widely used on an industrial level by using malt from barley (raw material in beer technology) produced by a gentle drying in order to preserve it enzymatic activity. The idea of our research was to identify and further develop other ingredients (in our case germinated soybean flour) which avoid or at least reduce the use of replacers like additives in wheat flour by achieving natural and good bakery products from the nutritional and technological point of view. In our study the wheat flour does not contain additives and is with a low alpha amylase activity (has a high falling number values). Thorugh germination the soybean enzymatic activity increased (we reported the falling number value descreased in other previous research) which we mentioned it in our manuscript (see: https://www.mdpi.com/2076-3417/11/24/11706) We did some experiments and with no germinations the falling number of mix flours between soy and wheat does not decreased very much. So, the germinated soybean flour is a completly different ingredient that the native soybean one which besides an ingredient that improves bread nutritional valus it also improves it technological characteristics due to it enzymatic activity (of course for wheat flours  with a high falling number values).

Referee comments: Line 186-192. Explain in text why for GSF increased up 17.9% and 40.2% fat and protein respectively.

Response: We want to tahnk to the referee for the close reading of our manuscript. The 17.9% and 40.2% are the values for germinated soybean flour for fat and protein. Wheat flour is a refined one which present a much lower protein and fat content. We explained in a more extensive way now why is the difference. I think was a confusion due to the English language (it is not an increased are the GSF data). We have reviewed the formulation.

Referee comments: Line 191. Add  the characteristics for soybean flour without germination. Discuss in text.

Response: We added the characteristics for soybean flour without germination and explained the reasons behind the changes that occur in the characteristics of soybean flour due to germination.

Referee comments: Line 221.  “microsturctures”.

Response: We have modified the form of the word in the manuscript. Thank you very much for pointing us this mistake.

Referee comments: Line 219. 3.3. Dough microstructure.   Comment: This technique is optical and does not measure microstructure. Discuss in text why this technique is considered a microstructural test.

Response: We completed and discussed in our manuscript that EFLM is an optical one and why is prefered for microstructural analysis. We have reported in particular to the article published by – see reference 35 :

https://www.sciencedirect.com/science/article/abs/pii/S0733521009001386

Referee comments:Fig 1. According with image the red color is protein (rhodamine) . Why if you increase 20% of GSF, there are as 50% of increases in red color (Fig 1.e) . Discuss in text.

Response: We have now discussed in the manuscript  in a more extensive way why this changes occured.

Referee comments: Note: Rhodamine is a marker colorant that is not approved by FDA.

Response: We agree with the referee point of view. In the research studies it is the most used fluorochromes for dough microstructure.

Referee comments: Line 290. Fig 3. Why control has as poor values? I think that control must have the highest values and the mixes reaching those values. Isnt it?. Discuss in text

Response: We believe that this was becouse that we used a wheat flour with no additivation which was a not very good one for bread making (low alpha amylase activity), of strong quality for bread making (a gluten deformation index of 3 mm). So, using GSF (which has an enzymatic activity) the gluten extension was improved and also the amylase activity and carbon dioxide formed. More, GSF brings some lipids which positively affect bread quality. As it may be seen the control sample has low values for loaf volume, porosity, elasticity compared to the samples with GSF addition which was also reflected in bread sensory quality. We explained in the discussion section the reasons that led to the better appreciation of the samples with 5%, 10% and 15% addition of GSF than the control sample by the participants of the sensory session.

Referee comments: Line 282. I cannot see that GSF addition increase pore size. I see in fig 2 that pore  size reduce as GSF increase.  So , I suggest to author that measure the size pore for each plot of figure 2 and add a table showing size and discuss it.

Response: We want to thank to the referee for his/her suggetions. We completed our manuscript with table 6 – computed bread crumb characteristics which we discussed in the manuscript.

Referee comments: Line 486-489 . Author must prove the improved quality of the mixes as Rizzello et al proved it. How measure quality bread? Include in materials and method how to measure the quality bread. So, author can include in conclusion that quality bread was improved.

Response: We want to thank to the referee for his/her suggetions. We completed our manuscript with table 7 Compositional analysis of bread samples with different levels of germinated soybean flour (SGF) addition which we discussed in the manuscript.

Referee comments: Conclusion: In general, mixes increasing the firmness and this means that mixes flour were more hard and I think that this parameter not improve the quality bread. Why author conclude that the decrease of firmness is better for the quality bread?

Response: We agree with the referee point of view. The firmeness parameter does not indicates an improvement of bread quality. However, the other characteristics such as specific volume, porosity, elasticity, sensorial ones, compositional characteristics of bread samples, e.g. make us to conclude that the addition of GSF had an overall effect of improving the bread quality.

Referee comments: References: Add in reference list this article: Foods 2021, 10, 1542. https://doi.org/10.3390/foods10071542.

Response:We have added the recommended article in the reference list.

Finally, the authors would like to thank to the reviewer for his/her appreciations and for all the suggestions because these helped us to correct our paper and to optimize it.

Sincerely,

Georgiana Codină et al.

Reviewer 2 Report

The manuscript entitled "Dough rheological behavior and bread quality as affected by addition of germinated soybean flour" aimed to investigated the effect of germinated soybean flour on the dough and bread quality. This is interesting and could provide a valuable information for the application of GSF on breadmaking. Some revisions should be made as follows:

(1)  L110: Did the dough contain yeast during rheology test?

(2)  L194: please show the figure of creep test.

(3)  L361: During germination, the protein solubility, foaming capacity and the starch gelatinization has changed. Have you tested the exact changes of starch, protein and fat of soybean?

(4)  L375: The paper mentioned that the GSF could increase the capacity of the dough network to produce CO2, why and how?

(5)  Also, after germination, the a-amylase enzyme has activated and the amount of reducing sugars and amino acids increased? So, how much is the activity of a-amylase enzyme in GSF, how much has the content of amino acid and reducing sugar increased after germination? What is the composition and molecular weight of starch and protein? How the starch hydrolysis in dough with the addition of GSF?

(6)  The explanation of the change of bread quality needs to be confirmed by more mechanical experiments.

Author Response

16 March 2023

Dear Referee,  

We would like to thank the referee for the close reading and for the proper suggestions. We hope that we provide all the answers to the reviewer’s comments.

Thank you very much for the recommendations to publish our paper entitled “Dough Rheological Behavior and Bread Quality as Affected by Addition of Germinated Soybean Flour”.

The present version of the paper has been revised according to the reviewer’s suggestions.             

We uploaded the corrected version of the article for which we used the red color for the addition text.

Referee comments: The manuscript entitled "Dough rheological behavior and bread quality as affected by addition of germinated soybean flour" aimed to investigated the effect of germinated soybean flour on the dough and bread quality. This is interesting and could provide a valuable information for the application of GSF on breadmaking. Some revisions should be made as follows:

Response: We want to thank to the referee for the close reading and his/her appreciations. We hope that our revised form to be an improved one, sustainable for publication.

Referee comments:  (1)  L110: Did the dough contain yeast during rheology test?

Response: No, the dough does not contain yeast. In general, the dynamic tests are done without the addition of yeasts because the device is very sensitive, the dough would rise and problems would appear in the analysis. We mentioned now in the text that the dough does not contain yeast, in Line 126.

Referee comments:  . (2)  L194: please show the figure of creep test.

Response: We would like to thank to the referee for his/her suggestions. We added now in the manuscript  the figure for creep test (see Figure 1 between line 248-249).

Referee comments: (3)  L361: During germination, the protein solubility, foaming capacity and the starch gelatinization has changed. Have you tested the exact changes of starch, protein and fat of soybean?

Response: We did not do exact tests to monitor the changes of starch, protein and fat structure and technological characteristics of soybean that occurred due to germination. We have made these comments and explanations based on the results reached by other researchers in their previous studies. However, we completed our manuscript with chemical characteristics of composite flours with different proportions of GSF supplementation (table 1 – lines 231-244) in which we reported the protein, fat, ash and carbohydrates variation.

Referee comments: L375: The paper mentioned that the GSF could increase the capacity of the dough network to produce CO2, why and how?

Response: We want to thanks to the referee for the close reading of our manuscript. We completed our manuscript with more explanations such as GSF addition decreased the falling number value and therefore the alpha amylase activity which will conduct to a higher carbon dioxide production. We specified this in Lines 487-491.

Referee comments: Also, after germination, the a-amylase enzyme has activated and the amount of reducing sugars and amino acids increased? So, how much is the activity of a-amylase enzyme in GSF, how much has the content of amino acid and reducing sugar increased after germination? What is the composition and molecular weight of starch and protein? How the starch hydrolysis in dough with the addition of GSF?

Response: We completed our revised manuscript with more information’s related the changes that may occur by GSF addition in wheat flour. The alpha-amylase activity increase by GSF addition was reported by us in a previous study which we mentioned it in this manuscript (https://doi.org/10.3390/app112411706) (see Lines 487-491).We did not determined the amino acids content of dough and reducing sugars by GSF addition in wheat flour but we previously reported the increased amount of total CO2 volume production to Rheofermentometer device whih may be direct correlated with reducing sugars increased. 

Referee comments: The explanation of the change of bread quality needs to be confirmed by more mechanical experiments.

Response: We want to thank to the referee for his/her suggestions. We made the loaf volume, porosity, elasticity, textural and sensory analysis which are the most important aspects from the technological point of view for the consumers. But we completed our manuscript with more data related to chemical composition of bread by GSF addition in it recipe (table 7). We also discussed the data obtained, in Lines 636-652.

Finally, the authors would like to thank to the reviewer for his/her appreciation and for all the suggestions because these helped us to correct our paper and to optimize it.

Sincerely,

Georgiana Codină et al.
